# Effects of Decabrominated Diphenyl Ether Exposure on Growth, Meat Characteristics and Blood Profiles in Broilers

**DOI:** 10.3390/ani11020565

**Published:** 2021-02-22

**Authors:** Zehui Liu, Hulong Lei, Renyong Tang, Junhua Yang, Xiulan Guo, Renmao Huang, Qinxiong Rao, Lin Cheng, Zhihui Zhao

**Affiliations:** 1Institute for Agro-Food Standards and Testing Technology, Shanghai Academy of Agricultural Sciences, Shanghai 201403, China; liuzehui33@163.com (Z.L.); yangjunhua303@126.com (J.Y.); hrenmao@163.com (R.H.); qinxiongrao@163.com (Q.R.); chenglin_8813@126.com (L.C.); 2Institute of Animal Husbandry and Veterinary Sciences, Shanghai Academy of Agricultural Sciences, Shanghai 201106, China; leihulong@saas.sh.cn; 3School of Pharmacy and Biological Engineering, Chengdu University, Chengdu 610106, China; tangrenyong@cdu.edu.cn (R.T.); guoxiulan@cdu.edu.cn (X.G.)

**Keywords:** decabrominated diphenyl ether, toxicity assessment, meat characteristics, antioxidant capacity, broilers

## Abstract

**Simple Summary:**

Decabrominated diphenyl ether (BDE-209) has been widely used in recent decades, and its cumulative toxicity to crops, animals and human beings is attracting increasing attention. In this study, we established broiler models to investigate the effects of BDE-209 exposure at doses of 0, 0.02, 0.4 and 4 mg/kg on growth, carcass traits, meat quality, blood profiles and antioxidant capacity. Our results suggested that BDE-209 exposure altered the blood profiles and carcass traits and exhibited toxicity in liver and kidney functions. Furthermore, BDE-209 increased plasma malondialdehyde (MDA) concentrations and decreased the activities of glutathione peroxidase (GSH-px) and superoxide dismutase (SOD), which implied aggravating oxidant stress and decline of antioxidant capacity in broilers.

**Abstract:**

Decabrominated diphenyl ether (BDE-209) is widely used as a flame retardant and is detected at high levels in the environment. Its toxicities have been reported and have attracted attention. In the present study, broilers were used to determine the response in growth performance, carcass traits, meat quality, blood profiles and antioxidant system to BDE-209 exposure at doses of 0, 0.02, 0.4 and 4 mg/kg. The results showed that BDE-209 exposure at levels of 0.02 or 0.4 mg/kg increased feed intake and decreased feed efficiency. BDE-209 altered the blood profiles, such as reducing the numbers of white blood cells, lymphocytes and neutrophilic granulocytes. As compared with the control, BDE-209 exposure significantly increased abdominal fat percentages of broilers at 64.9–159.5% and adversely affected the selected biochemical indicators, including alkaline phosphatase (ALP), aspartate aminotransferase (AST), alanine aminotransferase (ALT), creatine (CRE), which indicated its toxicity to liver and kidney functions. Moreover, BDE-209 exposure significantly increased plasma malondialdehyde (MDA) concentrations and decreased the activities of glutathione peroxidase (GSH-px) and superoxide dismutase (SOD), which implied aggravating oxidant stress and decline of antioxidant capacity in broilers. In conclusion, our data demonstrated that the environmental pollutant BDE-209 adversely influenced growth performance, increased the deposition of abdominal fat, impaired antioxidant capacity and the immune system and had potential toxicity to the liver and kidney of broilers.

## 1. Introduction

Polybrominated diphenyl ethers (PBDEs) are persistent organic pollutants (POPs) that are widely used as flame retardants in textiles, electronic appliances, construction materials. Their release, migration and accumulation in the environment during manufacture and usage contribute to the ubiquity of PBDEs. Around electronic waste recycling sites, contaminant PBDEs have been detected at high levels in the environment [1], feeds [2], fish [3], chicken [4], etc. Decabrominated diphenyl ether (BDE-209), as the predominant PBDE congener, has been detected in large quantities in sediments [5], dust [6], plants [7] and chickens [8,9] in polluted sites.

The occurrence of PBDEs is ubiquitous in various foods, animal feeds and raw materials, such as fish meal, soybean, rapeseed [2]. However, the concentrations of PBDEs vary greatly, not only in different foods and animal feeds but also in different countries and regions. The total concentrations of 17 measured PBDE congeners ranged from 0.02 to 8.91 ng/g whole weight for foods (fish, eggs, milk, meat, etc.), and from 0.11 to 9.63 ng/g whole weight for animal feeds [10]. BDE-209 was the most abundant congener in chicken feeds, accounting for more than 80% in the total concentrations of eight detected PBDEs [2]. BDE-209 concentration in liver and breast meat was up to 3.89 and 3.55 mg/kg, respectively, when chickens were fed with BDE-209 at 85 mg/kg for 60 days [11]. PBDEs can easily leach out of the contaminated materials, accumulate in the environment and migrate through the food chain, which is a serious threat to the health of human beings, who are at the top of the chain. The concentrations of PBDEs in human serum have been detected and have varied greatly among regions and countries, including European countries [12], the United States [13] and China [14]. Higher concentrations of PBDEs in serum have been attributed to contaminated food and occupational exposure in the workplace during manufacture or recycling of electronic products. It has been reported that the average concentration of PBDEs in 3–5 year old infants was 3.4 times higher than in 12–19 year old adolescents, although there was no change in adults with increasing age [13].

BDE-209 exhibited toxicity in thyroid [15], liver [16], nervous [17] and reproductive systems [18] of mice. Dietary exposure is an important condition for BDE-209 bioaccumulation and subsequent bio-magnification throughout food chains, which are attributed to their lipid solubility and refractory properties [8,19]. BDE-209 has been reported at high levels in fishmeal, which is used widely in the global animal feedstuff industry. Animal feeds are considered to be the beginning of the farm-to-fork pathway [2]. However, information about the biological toxicity of dietary BDE-209 exposure and its sensitive indicators is limited. 

Poultry production remains at the top of global meat production and plays an important role in meeting the increasing demands for safe and healthy meat products. Diet is the main pathway for exposure to PBDEs, both in humans and animals. In the present study, an animal model was established to investigate the potential hazards of BDE-209 at different levels on growth performance, carcass traits, meat quality and blood profiles in chicken. Our study also seeks to help better understand the health risk of dietary cumulative BDE-209 for the special population consuming PBDE contaminated foods.

## 2. Materials and Methods

### 2.1. Animals

A total of 80 one-day-old male Arbor Acres broilers were chosen, and they were raised in stainless steel cages, receiving water and feed *ad libitum*. Birds were fed with corn–soybean diets at different phases, which were formulated to meet or exceed Nutrient Requirements of Poultry recommended by National Research Council of the National Academies (NRC, 1994). Diets were analyzed for crude protein (CP), calcium (Ca) and phosphorus (P) following Association of Official Analytical Chemists (AOAC, 2000) procedures, and the nutritional composition is listed in Table 1. All experimental cages were in the same room, and the temperature was maintained at approximately 34 °C during the first 3 days and was gradually reduced to 28 °C by the end of week 2. Artificial light was provided for 20 h/d via fluorescent lights.

Animal care and use procedures were followed in compliance with the Guidelines of the Animal Ethics Committee of Shanghai Academy of Agricultural Sciences, in accordance with standard international regulations. The experimental protocols used for research purposes in the present study were approved by the Institutional Review Board (SAASPZ0920001).

### 2.2. Experimental Design

To investigate the potential hazards of BDE-209, 80 male broilers were randomly allocated into 4 groups according to their body weights after feeding with the basal corn–soybean diet for one week, with 5 replicate cages of 4 broilers per cage. The control group was fed with the basal diets, and the BDE-209-treated groups were fed with the basal diets supplemented with 0.02, 0.4 or 4 mg/kg BDE-209 (Macklin, Shanghai, China, purity ≥ 98%), which was uniformly premixed with basal diet meal powder and incorporated step by step. Broilers were raised under the same conditions and management except for diet, with different doses of BDE-209 throughout the experiment. The trial lasted for 6 weeks and was divided into two phases, namely weeks 1–3 and 4–6. In the different phases, chickens were fed with different diets based on their nutritional requirements, but the levels of supplemental BDE-209 in their diets were unchanged. 

### 2.3. Growth Performance

Body weight and feed consumption of each replicate cage were weighed every week at 7 a.m. after 12 h of starvation. Average daily gain (ADG) and average feed intake (ADFI) as well as the feed conversion ratio (FCR, feed/gain) were calculated according to these data. The number and weight of dead broilers were recorded, and the mortality rate was calculated.

### 2.4. Carcass Characteristics and Meat Quality

At the end of the experiment, birds from each group were weighed individually and sacrificed by cervical dislocation and bleeding via jugular vein. Carcasses were weighed prior to being skinned and deboned. Breast muscle, thigh muscle and abdominal fat were removed, weighed and expressed as a percentage of body weight. Breast muscle from the right side was used to determine the carcass characteristics. The breast meat lightness (L*), redness (a*) and yellowness (b*) values were measured with an SP64 color-difference meter (X-Rite, Grand Rapids, MI, USA), and pH values were detected at 45 min postmortem using an Orion Star pH meter (Thermo Fisher Scientific, Waltham, MA, USA). Drip loss was measured using approximately 5 g of meat sample according to the plastic bag method described by Honikel (1998) [20], and shear force values of breast muscle were measured using an RH-N50 meat tenderness tester (Mingao, Nanjing, China).

### 2.5. Blood Profiles

At the end of week 6, all broilers were bled via the wing vein using a sterilized syringe after 12 h of fasting. Blood samples were collected in vacuum tubes with heparin. The whole blood samples were immediately used to detect the blood parameters of broilers using an automatic hematology analyzer (Sinnowa, Nanjing, China), including numbers of white blood cells (WBCs), lymphocytes (LYMs), intermediate cells (MIDs), neutrophilic granulocytes (NEUs), red blood cells (RBCs) and platelets (PLTs), as well as hemoglobin content (HGB), hematocrit (HCT), platelet distribution width (PDW) and plateletcrit (PCT).

Plasma was separated by centrifuging at 4000× *g* for 15 min at 4 °C and stored at −80 °C for subsequent biochemical analysis. Plasma was subsequently used to determine the selected biochemical indicators, using an automatic biochemistry analyzer (Beckman Coulter, Brea, CA, USA), including aspartate aminotransferase (AST), alanine aminotransferase (ALT), gamma glutamyl transferase (GGT), alkaline phosphatase (ALP), albumin (ALB), glucose (GLU), serum urea (UREA), total bilirubin (TBIL), creatinine (CRE) and uric acid (UA). All sample analyses were performed in duplicate for the determination of blood parameters and plasma biochemical indicators.

### 2.6. Antioxidant Capacity

According to the manufacturer’s directions, plasma samples were used to determine malondialdehyde (MDA) concentration, the activities of glutathione peroxidase (GSH-Px) and superoxide dismutase (SOD) and total antioxidant capacity (T-AOC) with the analysis kits (Nanjing Jiancheng Bioengineering Institute, Nanjing, China). All samples and standards were performed in triplicate.

### 2.7. BDE-209 Content in Breast Muscle

Samples were extracted with N-hexane/dichloromethane (1:1, *v*/*v*) solvent and purified by acid silica gel column. Extracts were analyzed with an Agilent 6890A gas chromatograph with an electron capture detector (ECD). A DB-5HT capillary column (30 m length, 0.32 mm i.d.; 0.25 μm film thickness) was used to separate the target analyte. The column temperature was programmed from 140 (held for 2.0 min) to 180 °C at 5 °C/min (held for 5.0 min), increased to 265 °C at 5 °C/min (held for 5.0 min) and finally elevated to 315 °C at 15 °C/min (held for 10.0 min). One microliter of each extract was injected automatically in the splitless mode. Ultrahigh purity helium served as the carrier gas at a flow rate of 2.0 mL/min. 

The standard substance of PBDE-209 was purchased from AccuStandard Inc. (New Haven, CT, USA). The reporting limit for BDE-209 was 0.125 ng/g for 20 g wet sample weight (defined as signal-to-noise ratio (S/N) = 3), and the method limit of quantification (defined as signal-to-noise ratio (S/N) = 10) of matrix samples was spiked at a concentration of 0.4 ng/g. The external standard method was used for accurate quantification. To guarantee repeatability, procedural blank matrix samples were spiked in duplicate and standard reference material for each batch of 20 samples. The recovery of the spiked samples was 70.0–120% at the level of 10 ng/g, and the relative percent difference was less than 20% between the two spiked samples.

### 2.8. Statistical Analysis

Data were analyzed with IBM SPSS Statistics 20.0 (IBM SPSS Statistics, Chicago, IL, USA). After confirming normal distribution on histograms, differences among samples with experimental treatments were evaluated by one-way ANOVA. The generalized linear models (GLM) procedure was performed using the method of least-significant difference (LSD) multiple comparisons. Regression analysis was performed by linear and quadratic methods to study the effects of incorporation of BDE-209 in different diets. A *p*-value of 0.05 was considered to be statistically significant.

## 3. Results

### 3.1. Growth Performance

The effects of dietary BDE-209 exposure on the growth performance of broilers are shown in Figure 1. A dose-independent decrease in average daily gain was detected (linear, *p* = 0.05), although there was no significant difference in ADG between the control and BDE-209-treated groups (Figure 1A, *p* > 0.05). As compared to the control group, dietary exposure of BDE-209 at the levels of 0.02 and 0.4 mg/kg increased both ADFI (Figure 1B, *p* = 0.00, *p* = 0.00) and ratios of feed to gain (F/G) in broilers (Figure 1C, *p* = 0.01, *p* = 0.00). Dietary exposure of 4 mg/kg BDE-209 reduced ADFI (*p* = 0.01) and ADG (*p* = 0.10), but no change was observed in feed efficiency of broilers (F/G) (*p* = 0.93). There was only one dead chicken in the control and BDE-209-treated group at the dose of 0.4 mg/kg throughout the 6-week experiment, and it was unrelated to BDE-209 treatment. 

### 3.2. Carcass Traits and Meat Quality

As shown in Table 2, dietary BDE-209 exposure exerted no influence on dressing percentage and eviscerating percentage (*p* > 0.05). As compared to the control, 0.02, 0.4 and 4 mg/kg BDE-209 decreased breast muscle percentage by 6.35% (*p* = 0.08), 2.41% (*p* = 0.46) and 6.15% (*p* = 0.11), and decreased thigh muscle percentage by 2.37% (*p* = 0.59), 6.86% (*p* = 0.07) and 5.61% (*p* = 0.18), respectively. Unexpectedly, BDE exposure, regardless of level, significantly increased abdominal fat percentage of broilers by 64.9–159.5% in comparison with the control (*p* < 0.01). Nevertheless, there was no linear (*p* = 0.25) or quadratic correlation (*p* = 0.28) between abdominal fat percentages and the doses of dietary BDE-209.

The effects of BDE-209 on breast pH, meat color, tenderness and drip loss are shown in Table 3. It was suggested that BDE-209 at the levels of 0.4 and 4 mg/kg significantly decreased the pH values of broiler breast (*p* = 0.05, *p* = 0.00). Furthermore, BDE-209 exposure increased a* values regardless of levels (*p* < 0.05) and decreased the L* value at 0.4 mg/kg compared with the control. Increasing the levels of BDE-209 exposure decreased the pH values of breast muscle in a dose-dependent manner (linear, *p* = 0.05) but did not affect breast shear force, drip loss or meat color (lightness, redness and yellowness) (linear, *p* > 0.05).

### 3.3. Blood Parameters

The effects of BDE-209 on blood parameters of broilers are presented in Table 4. As compared to the control, a significant decrease in the numbers of white blood cells (WBCs), lymphocytes (LYMs), intermediate cells (MIDs), neutrophilic granulocytes (NEUs) and platelets (PLTs) (*p* < 0.01) was determined in the groups with 0.02 or 0.4 mg/kg treatments. Moreover, dietary exposure with 0.02 and 0.4 mg/kg BDE-209 increased red blood cells (RBCs), hemoglobin (HGB), hematocrit (HCT) (*p* < 0.01) and platelet distribution width (PDW) (*p* < 0.05). Furthermore, our results suggested that dietary BDE-209 exposure at 0.4 mg/kg seemed to be the turning point, and treatment at 4 mg/kg alleviated or abolished the influence of BDE-209 in most of the selected blood parameters, except PLT and PCT.

### 3.4. Plasma Biochemical Indicators

The effects of BDE-209 on selected biochemical indicators of broilers are shown in Table 5. Compared with the control, 0.4 mg/kg BDE-209 increased plasma gamma glutamyl transferase activity (GGT) (*p* = 0.04) and albumin (ALB) concentrations (*p* = 0.03) and decreased creatinine concentration (CRE) (*p* = 0.00). Moreover, 4 mg/kg BDE-209 increased aspartate aminotransferase activity (AST) (*p* = 0.035), as well as the concentrations of plasma glucose (GLU) (*p* = 0.02) and urea (UREA) (*p* = 0.00). BDE-209, both at the levels of 0.4 and 4 mg/kg, significantly increased alanine aminotransferase activity (ALP) (*p* = 0.00 for both levels) and the concentrations of plasma total bilirubin (TBIL) (*p* = 0.00 for both) and uric acid (UA) (*p* = 0.01 for both) as compared to the control. As for plasma alkaline phosphatase activity (ALP), a significant decrease was detected in BDE-209-treated groups, regardless of levels, as compared with the control (*p* = 0.00, *p* = 0.00, *p* = 0.00, respectively). The results showed that the exposure of dietary BDE-209 with increasing doses elevated ALT (linear, *p* = 0.05; quadratic, *p* = 0.03).

### 3.5. Antioxidant Capacity

As shown in Figure 2, the results suggested that BDE-209 significantly increased plasma malondialdehyde (MDA) levels (Figure 2A), reduced total antioxidant capacity (T-AOC) (Figure 2B) and decreased the activities of glutathione peroxidase (GSH-px) (Figure 2C) and superoxide dismutase (SOD) (Figure 2D) as compared with the control group. Increasing the doses of BDE-209 exposure had no apparent effect on plasma MDA (linear, *p* = 0.74), T-AOC (linear, *p* = 0.36), GSH-px (linear, *p* = 0.54) and SOD (linear, *p* = 0.88).

### 3.6. Accumulation of BDE-209 in Breast Muscle

The concentrations of BDE-209 in breast muscle were determined by gas chromatography after a 6-week exposure of dietary BDE-209, and the results are shown in Figure 3. Results suggested that bioaccumulation of BDE-209 in breast muscle significantly increased in a dose-dependent manner as the doses of BDE-209 increased (linear, *p* = 0.00; quadratic, *p* = 0.00). The average BDE-209 concentrations in breast of treatment groups of 0.02 (92.84 ng/g lipid weight), 0.4 (165.01 ng/g lipid weight) and 4 mg/kg BDE-209 (235.38 ng/g lipid weight) were 39.72, 71.38 and 103.24 times higher than the concentration of the control group (2.28 ng/g lipid weight) (*p* < 0.01).

## 4. Discussion

As a brominated flame retardant, BDE-209 has been widely used in a variety of electronic and construction products, which has caused its release to the environment during manufacture and usage [3]. One of the most important ways that domestic animals are exposed to organic pollutants such as PBDEs is through the intake of contaminated diets. PBDE exhibits potential adverse effects on animals and humans due to its bioaccumulation in the food chain.

In a previous study, American kestrel (*Falco sparverius*) nestlings exposed to PBDE congeners (BDE-47, -99, -100 and -153) were reported to have more feed intake and body weight gain [21]. However, investigations in mice and *Xenopus tropicalis* (West African clawed frogs) showed that exposure to BDE-47 caused growth retardation [22,23], which implied that the inconsistency in retrospective studies might be attributed to differences in types of PBDE congeners, dosage and animal species. In this study, the doses of BDE-209 are much lower than the mortality dose in chickens. There was only one dead bird in each of the control and the 0.4 mg/kg of the BDE-209-treated group throughout the 6-week experiment. This mortality was unrelated to dietary BDE-209 exposure. There was a tendency towards a dose-dependent decrease in body weight gain observed in broilers exposed to BDE-209, which was consistent with [22,23]. BDE-209 exposure at 0.02 or 0.4 mg/kg increased feed intake and decreased feed efficiency. As for broilers exposed to 4 mg/kg BDE-209, the reductions both in feed consumption and body weight gain contributed to no change in feed conversion.

In the present study, dietary BDE-209 exposure exerted an influence on meat quality of chicken, but the changes of carcass traits, especially in abdominal fat percentage, deserve more attention in BDE treatments. Previous research indicated that adipose tissue should be considered as a target tissue for PBDEs [24], and PBDE exposure during critical windows of development might contribute to the increasing prevalence of obesity. Metabolomics investigations showed that PBDE exposure altered serum metabolites, which are primarily implicated in amino acid, lipid, carbohydrate and energy metabolisms [25]. Similarly, a significant increase in abdominal fat deposition in broilers was observed after oral BDE-209 treatment in the present study, which implied that lipid metabolism and fat accumulation of broilers could be affected by BDE-209 exposure. However, more deposition of abdominal fat is not only unfavorable in chicken consumption but is also detrimental to the health of chickens. For example, reproductive performance is better in lean birds than in fat birds, and abdominal fat deposition negatively affects the reproductive performance of birds [26,27]. 

A previous study indicated that plasma biochemical indicators, including AST, ALT, GGT, TBIL, CRE and UA, differed significantly between lean and fat animals and correlated significantly with abdominal fat [28]. Correspondingly, our data suggested that BDE-209 exposure increased AST, ALT, GGT, TBIL and UA, and decreased CRE and ALP. These could be related to an increase in abdominal fat deposition and liver and kidney functions in BDE-209 treatment. ALT is an important enzyme that is mostly produced by hepatocytes and that is usually confined to liver cells [29]. However, it can be released into the blood when the liver is undergoing stress, and increasing ALT is considered to be a sensitive indicator of liver damage and metabolic syndrome [30]. In this study, our results suggested that plasma ALT was increased by increasing amounts of dietary BDE-209 exposure. It is suggested that ALT could be a candidate marker of BDE-209 exposure, as liver tissue is the major target tissue of BDE-209 [31]. ALP comes primarily from liver, kidney and bone, and it plays an important role in bone development and the functions of the liver and kidney. In the present study, ALP activities of broilers were decreased over 30% by BDE-209 exposure, which indicated that BDE-209 exposure had a large influence on the functions of the liver and kidney. This was similar to the results in mice [16]. Moreover, the numbers of white blood cells, lymphocytes and neutrophilic granulocytes were decreased by BDE-209 exposure in this study, which indicated that the immune system may have been impaired. Compared with previous studies of toxicity assessment, our study focused on the potential hazards of accumulated BDE-209 in diets, and the doses used in the present study were much lower than mortality dose of BDE-209. The hematological and biochemical parameters varied with BDE-209 treatment within the range of normal physiological and zootechnical references.

Free radicals and reactive oxygen species are widely distributed in various tissues, and animals maintain complex systems to balance the oxidative state. As components of the antioxidant system, antioxidant enzymes, including GSH-px and SOD, play important roles in free radical scavenging. A previous study suggested that oral intake of BDE-209 increased the level of serum CRE and thiobarbituric acid reactive substances (TBARS), which indicated oxidative imbalance and nephrotoxicity in rat [32]. BDE-47 induced oxidative stress, DNA damage and apoptosis, accompanied with increasing MDA contents and decreasing GSH and SOD in a dose-related manner [33]. In this study, BDE-209 exposure consistently elevated plasma MDA concentrations and reduced the activities of GSH-px and SOD, which indicated the reduction in the antioxidant capacity of broilers. Further investigations are required to clarify the underlying mechanisms responsible for the effects of BDE-209 on abdominal fat accumulation, blood profiles and antioxidant capacity. 

## 5. Conclusions

The present study investigated the effects of the environment pollutant BDE-209 on the growth performance, carcass traits, blood profiles and antioxidant state of broilers. The results suggested that oral BDE-209 exposure decreased the body weight gain in a dose-dependent manner, elevated bioaccumulation in breast as the dose was increased and showed potential hazards in liver function and the antioxidant system in broilers. Further investigations will be required to understand the underlying toxic mechanism of BDE-209 both in human beings and animals.

## Figures and Tables

**Figure 1 animals-11-00565-f001:**
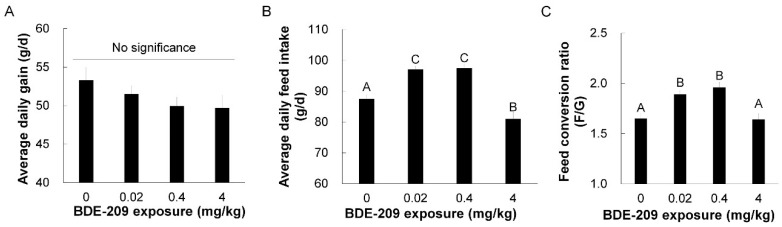
Growth performance of broilers exposed to BDE-209 at different doses. (**A**) Average daily gain (ADG); (**B**) average daily feed intake (ADFI); (**C**) feed conversion ratio (F/G). Bars represents mean values ± standard errors. Different large letters represent significant difference at *p* < 0.01.

**Figure 2 animals-11-00565-f002:**
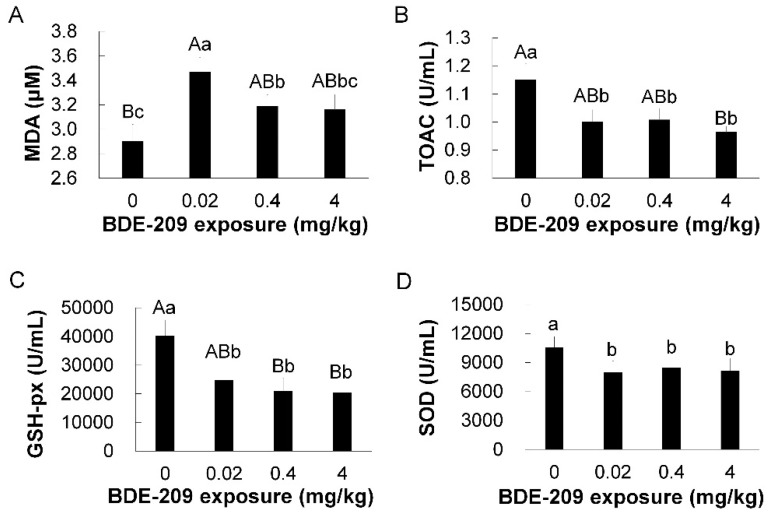
Antioxidant capacity of broilers exposed to BDE-209. Plasma samples were used to determine malondialdehyde (MDA, (**A**)) concentrations, total antioxidant capacity (T-AOC, (**B**)) and the activities of glutathione peroxidase (GSH-Px, (**C**)) and superoxide dismutase (SOD, (**D**)). Bars represent mean values ± standard errors. Different small letters represent significant difference at *p* < 0.05, and different large letters represent significant difference at *p* < 0.01.

**Figure 3 animals-11-00565-f003:**
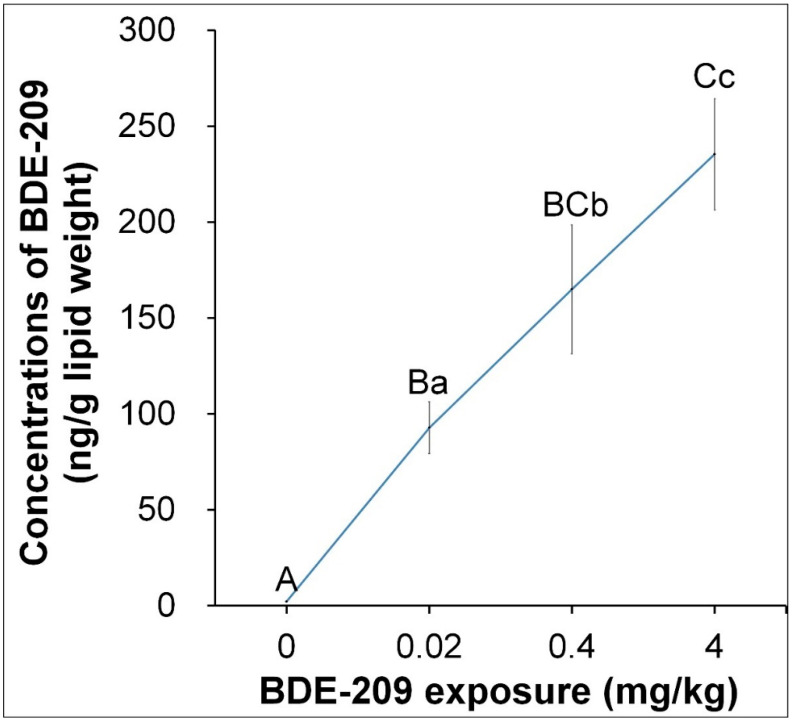
Concentrations of BDE-209 in breast muscle after exposure to BDE-209 at different doses for 6 weeks. Bars represents mean values ± standard errors. Different small letters represent significant difference at *p* < 0.05, and different large letters represent significant difference at *p* < 0.01.

**Table 1 animals-11-00565-t001:** Composition of the basal diets for broilers.

Composition of Diets	Weeks 1–3	Weeks 4–6
ME (calculated metabolizable energy, MJ/kg)	12.74	12.98
Crude protein (%)	21.36	19.6
Calcium (%)	1.02	0.89
Total phosphorus (%)	0.68	0.62
Lysine (%)	1.33	1.07

**Table 2 animals-11-00565-t002:** Carcass traits of broilers exposed to BDE-209 at different doses.

Items	Treatments of Dietary BDE-209 (mg/kg)					*p*-Value		
	0	0.02	0.4	4	SEM	ANOVA	Linear	Quadratic
Dressing percentage	90.38	91.06	90.46	90.13	0.72	0.43	0.14	0.98
Eviscerating percentage	88.07	88.27	88.87	88.31	1.78	0.81	0.55	0.22
Breast muscle percentage	14.96	14.01	14.60	14.04	0.52	0.21	0.25	0.33
Thigh muscle percentage	8.02	7.83	7.47	7.57	0.53	0.28	0.79	0.09
Abdominal fat percentage	0.37 ^A^	0.96 ^B^	1.02 ^B^	0.61 ^C^	0.25	0.00	0.25	0.28

Note: Different large letters represent significant difference among the means in the same row at *p* < 0.01.

**Table 3 animals-11-00565-t003:** Meat quality of chicken exposed to BDE-209 at different doses.

Items	Treatments of Dietary BDE-209 (mg/kg)					*p*-Value		
	0	0.02	0.4	4	SEM	ANOVA	Linear	Quadratic
Breast pH value	6.17 ^Aa^	6.13 ^Aa^	6.06 ^ABb^	5.95 ^B^	0.02	0.00	0.05	0.00
Lightness (L*)	48.76 ^a^	48.28 ^ab^	47.65 ^b^	48.61 ^ab^	0.20	0.19	0.10	0.12
Redness (a*)	1.80 ^A^	2.43 ^ABa^	2.52 ^Bab^	2.07 ^Bb^	0.08	0.00	0.24	0.05
Yellowness (b*)	15.09 ^AB^	15.87 ^A^	14.43 ^B^	14.26 ^B^	0.17	0.00	0.11	0.00
Breast shear force (N)	12.82	12.17	13.91	12.94	1.43	0.43	0.65	0.91
Drip loss	1.07 ^ab^	1.17 ^ab^	1.19 ^a^	0.97 ^b^	0.21	0.21	0.41	0.38

Note: Different small letters represent significant difference among the means in the same row at *p* < 0.05, and different large letters represent significant difference among the means in the same row at *p* < 0.01.

**Table 4 animals-11-00565-t004:** Changes in the blood parameters of broilers exposed to BDE-209.

Items	Treatments of Dietary BDE-209 (mg/kg)					*p*-Value		
	0	0.02	0.4	4	SEM	ANOVA	Linear	Quadratic
WBCs (10^9^/L)	9.88 ^Aa^	7.75 ^B^	6.46 ^C^	8.95 ^Ab^	0.23	0.00	0.29	0.00
LYMs (10^9^/L)	0.98 ^A^	0.82 ^B^	0.62 ^C^	1.04 ^A^	0.03	0.00	0.43	0.00
MIDs (10^9^/L)	0.31 ^A^	0.23 ^B^	0.20 ^C^	0.29 ^A^	0.01	0.00	0.81	0.00
NEUs (10^9^/L)	8.67 ^A^	6.76 ^B^	5.63 ^C^	6.71 ^B^	0.19	0.00	0.29	0.00
RBCs (10^12^/L)	2.41 ^C^	2.95 ^B^	3.27 ^A^	2.37 ^C^	0.06	0.00	0.93	0.00
HGB (g/L)	148.59 ^C^	170.18 ^B^	206.27 ^A^	131.42 ^D^	3.97	0.00	0.14	0.00
HCT (%)	27.58 ^C^	32.14 ^B^	37.98 ^A^	24.67 ^D^	0.73	0.00	0.43	0.00
PLTs (10^9^/L)	222.31 ^A^	183.18 ^Ba^	180.53 ^Bab^	167.06 ^Bb^	3.67	0.00	0.78	0.12
PDW (fL)	13.89 ^Bb^	18.83A ^ABa^	19.87 ^Aa^	18.56 ^ABa^	0.33	0.00	0.85	0.23
PCT (%)	0.23 ^A^	0.19 ^B^	0.20 ^B^	0.16 ^C^	0.00	0.00	0.53	0.03

Abbreviations: WBCs, white blood cells; LYMs, lymphocytes; MIDs, intermediate cells; NEUs, neutrophilic granulocytes; RBCs, red blood cells; HGB, hemoglobin; HCT, hematocrit; PLTs, platelet; PDW, platelet distribution width; PCT, plateletcrit. Note: Different small letters represent significant difference among the means in the same row at *p* < 0.05, and different large letters represent significant difference among the means in the same row at *p* < 0.01.

**Table 5 animals-11-00565-t005:** Plasma biochemical indicators of broilers exposed to BDE-209.

Items	Treatments of Dietary BDE-209 (mg/kg)					*p*-Value		
	0	0.02	0.4	4	SEM	ANOVA	Linear	Quadratic
AST (U/L)	275.83 ^a^	378.65 ^ab^	367.01 ^ab^	381.01 ^b^	17.65	0.12	0.13	0.20
ALT (U/L)	1.73 ^Bb^	2.18 ^ABb^	2.52 ^Aa^	2.80 ^Aa^	0.11	0.00	0.05	0.03
GGT (U/L)	25.99 ^ABb^	23.56 ^Bb^	31.29 ^Aa^	27.86 ^ABab^	1.02	0.05	0.31	0.71
ALP (U/L)	6861.08 ^A^	4281.09 ^B^	4680.49 ^B^	4692.63 ^B^	308.83	0.00	0.45	0.73
ALB (g/L)	11.12 ^ABb^	9.80 ^Bb^	13.12 ^Aa^	11.76 ^ABab^	0.38	0.01	0.30	0.24
GLU (mM)	14.7 ^ABb^	12.12 ^C^	14.34 ^Bb^	16.33 ^Aa^	0.33	0.00	0.11	0.36
UREA (mM)	0.82 ^A^	0.84 ^A^	0.86 ^A^	0.99 ^B^	0.02	0.00	0.21	0.11
TBIL (μM)	1.86 ^Bb^	2.14 ^Bb^	2.61 ^Aa^	2.47 ^ABa^	0.07	0.00	0.20	0.55
CRE (μM)	4.09 ^A^	2.33 ^B^	2.35 ^B^	4.77 ^A^	0.27	0.00	0.21	0.85
UA (μM)	205.08 ^Bb^	216.9 ^ABb^	278.23 ^Aa^	274.19 ^Aa^	10.15	0.01	0.32	0.10

Abbreviations: AST, aspartate aminotransferase; ALT, alanine aminotransferase; GGT, gamma glutamyl transferase; ALP, alkaline phosphatase; ALB, albumin; GLU, glucose; UREA, serum urea; TBIL, total bilirubin; CRE, creatinine; UA, uric acid. Note: Different small letters represent significant difference among the means in the same row at *p* < 0.05, and different large letters represent significant difference among the means in the same row at *p* < 0.01.

## Data Availability

The datasets used and/or analyzed during the current study are available from the corresponding author on reasonable request.

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
