# Peer review of "Effects of Decabrominated Diphenyl Ether Exposure on Growth, Meat Characteristics and Blood Profiles in Broilers"

_animals, 2021, doi:10.3390/ani11020565_

Round 1

Reviewer 1 Report

This manuscript is either confusing and of low interest or the authors did not justify properly why is this relevant to chicken health or production? Methods used in the manuscript are OK but I just cannot see why was this done? Every single section of the paper is too brief and vague. The introduction has to sell the story why we need to investigate this and why in chicken?

  • This is a broiler paper addressing the question that has little to no relevance to the chicken industry
  • Why use chicken as a model for BDE09 toxicity? This should be justified in introduction
  • How was the dose of BDE-209 selected? How do we know that dose has production or real relevance? How does this BDE 209 interfere with poultry production or health?
  • The experimental design stats with “To assess the toxicity of BDE-209” but toxicity was not fully evaluated, for example histology of major organs would be logical and a few classic tox analysis, instead the authors measured the meat quality, blood profile and anti-oxidants which is good but does not answer the exp. question.
  • The authors presume that the readers are experts in BDE209
  • Did you detect residual BDE-209 in blood or muscle?
  • “used as flame retardants in textiles, electronic appliances, construction materials, and so on” what does this have to do with chicken?
  • Introduction has ~300 words – not nearly enough
  • How does BDE get into chicken? Does it accumulate in meat? Does it get into humans when eating chicken
  • How do you get performance data on 20 broilers in each group???
  • Methods are vague and discussion needs to be expanded. 

Author Response

Comment:

This manuscript is either confusing and of low interest or the authors did not justify properly why is this relevant to chicken health or production? Methods used in the manuscript are OK but I just cannot see why was this done? Every single section of the paper is too brief and vague. The introduction has to sell the story why we need to investigate this and why in chicken? This is a broiler paper addressing the question that has little to no relevance to the chicken industry. Why use chicken as a model for BDE09 toxicity? This should be justified in introduction

Response:

Thanks for your comments and good suggestion! Sections of previous manuscript were too brief and vague, we have edited all the sections of our manuscript according to the comments of reviewers. Please review the revised manuscript.

As persistent organic pollutants (POPs), PBDEs has been used in several fields and they are ubiquitous not only in various agricultural products and processed foods, but also in animal feeds, raw materials (fish meal, soybean, rapeseed, etc.) and animal products as well. The concentration of BDE-209 is the highest in animal feeds and raw materials.

Poultry production remains at the top of worldwide meat production (just right after pig industry) and plays an important role in meeting the increasing demands for safe and healthy meat products. Besides, chicken is a good bioreactor and animal model for the studies of bioaccumulation and concentration. More and more attentions will be attracted on the hazards of cumulated BDE-209 in foods and animal feeds, because human beings are at the top of food chain.

Comment:

How was the dose of BDE-209 selected? How do we know that dose has production or real relevance? How does this BDE 209 interfere with poultry production or health?

The experimental design stats with “To assess the toxicity of BDE-209” but toxicity was not fully evaluated, for example histology of major organs would be logical and a few classic tox analysis, instead the authors measured the meat quality, blood profile and anti-oxidants which is good but does not answer the exp. question. The authors presume that the readers are experts in BDE209

Response:

Thanks for your comments. BDE-209 has been used in several fields. It can be leach out from these contaminated products. The release, migration and accumulation of BDE-209 in the environment during manufacture and usage contribute to the ubiquity of PBDEs. Previous retroactive studies suggested that BDE-209 has been detected in the commercial feeds of livestock (including pigs, chickens, ducks, fish, etc.) and raw materials (including fish meal, soybean, rapeseeds, etc.) though the concentrations are very low and varied greatly (ranged from 0.3-20 ng/g for common animal feeds and raw materials).

The tolerance to BDE-209 of animals varied greatly and the mortality dose is much higher than those used in the studies. Higher doses in diets almost were chosen (ranged from 1-85 mg/kg) in previous studies for toxicity assessment. In this study, the doses (0.02-4 mg/kg) was chosen according to the contents of some raw materials in previous reports and our preliminary experiment.

The comments are right that our study is not a classic study for the toxicity assess of BDE-209. As description above, the doses used in our study are much lower than previous studies. This study was conducted to investigate the potential hazards of BDE-209 on growth, meat quality and blood profiles (hematological, biochemical and anti-oxidant parameters) in chicken. Therefore, in the revised manuscript, we used the word “effects” instead of “toxicity assessment” in the title.

Comment: Did you detect residual BDE-209 in blood or muscle? “used as flame retardants in textiles, electronic appliances, construction materials, and so on” what does this have to do with chicken? Introduction has ~300 words – not nearly enough. How does BDE get into chicken? Does it accumulate in meat? Does it get into humans when eating chicken

Response:

We have determined the concentrations of accumulated BDE-209 in breast muscle but not blood, and supplemented the data in the revised manuscript.

BDE-209 can be leach out from these contaminated products. It was bioaccumulated and concentrated in aquatic and agricultural products, as well as transferring via food chain. Its release, migration and accumulation in the environment during manufacture and usage contribute to the ubiquity of PBDEs. There is a potential risk for the populations consumed enough contaminated foods (especial for seafoods) and occupational worker in electronic manufacture and recycle. Our study is also helpful to understand the potential hazards of dietary cumulative BDE-209 better.

Comment:

How do you get performance data on 20 broilers in each group???

Methods are vague and discussion needs to be expanded.

Response:

    Thanks for your comment and good suggestion. We have expanded and edited the section of methods according to the comment in the revised manuscript. In this study, 80 male broilers were randomly allocated into 4 groups according to their body weights after feeding with the basal corn-soybean diet for one week, 20 broilers in each group, 5 replicate cages of 4 broilers per cage.

Reviewer 2 Report

Dear Authors,

The subject of the article " Toxicity Assessment of Decabrominated Diphenyl Ether Exposure on Growth, Meat Characteristics and Blood Profiles in Broilers" is interesting, but does not seem publishable in its current form. Statistical analysis should give more interest to this work. The  bibliography  should be enriched.

Specific comments:

-The introduction should be enriched: justify the use of broiler as an animal model in your study. Also, look in the scientific literature for the real impact of Decabrominated Diphenyl Ether Exposure on animals and livestock products. It would be useful to further develop the interest and objectives of your study.

-In Material and Methods: What technique did you use to mix small amounts of Decabrominated Diphenyl Ether (0.02, 0.04 and 4 mg) per kg of feed? Why did you choose these incorporation rates?
Explain the sex ratio of your chickens?
Were there any mortalities during your experiment?

-For statistical analyses, since you data for monitoring growth and feed intake follow a normal distribution it would be more interesting to use a statistical model (General linear model or Mixed if you have individually identified the chickens) to take into account for these growth variables the effects of age and sex. Similarly for the other parameters studied, take into account the effect of sex as it is a factor that can significantly influence the results.

-Discussion did not flow with data presented:

It is necessary to explain the results observed and to check if the observed values are normal (physiological and zootechnical) or not.
Also give the mortality rates and explain the degree of toxicity of the supplemented feeds.

-Conclusion: Adapt your conclusion to the results of mortality and sex ratio by group.

Author Response

Comment:

The subject of the article " Toxicity Assessment of Decabrominated Diphenyl Ether Exposure on Growth, Meat Characteristics and Blood Profiles in Broilers" is interesting, but does not seem publishable in its current form. Statistical analysis should give more interest to this work. The bibliography should be enriched.

Response:

    Thanks for your comments and good suggestion. According to the comments, we have edited our manuscript, performed a regression analysis as well GLM using the method of least-significant difference (LSD) multiple comparisons. And the bibliography has been enriched.

Specific comments:

Comment:

The introduction should be enriched: justify the use of broiler as an animal model in your study. Also, look in the scientific literature for the real impact of Decabrominated Diphenyl Ether Exposure on animals and livestock products. It would be useful to further develop the interest and objectives of your study.

Response:

    Thanks for your comment and the suggestion is very good and helpful. The section of introduction was too brief and confusing in our previous manuscript. We have edited this section after referenced some scientific literatures.

Comment:

In Material and Methods: What technique did you use to mix small amounts of Decabrominated Diphenyl Ether (0.02, 0.04 and 4 mg) per kg of feed? Why did you choose these incorporation rates? Explain the sex ratio of your chickens? Were there any mortalities during your experiment?

Response:

Thanks for your comment, please see the revised manuscript. Our description in the previous manuscript is not detailed and clear. In this study, all the basal diets used both in the control and the experimental groups is the same batch production to avoid minor difference in nutritional levels during mixture. BDE-209 was uniformly premixed with basal diet meal powder and incorporated step by step.

The description in Material and Methods was vague. As for the toxicity of PBDEs, more chickens could not be raised for the present investigation, we just chosen 80 male broilers and divided into 4 groups to avoid the interference of sex on the results.

Previous retroactive studies suggested that BDE-209 has been detected in the commercial feeds of livestock (including pigs, chickens, ducks, fish, etc.) and raw materials (including fish meal, soybean, rapeseeds, etc.) though the concentrations are very low and varied greatly (ranged from 0.3-20 ng/g for common animal feeds and raw materials). Higher doses in diets almost were chosen (ranged from 1-85 mg/kg) in previous studies for toxicity assessment. In this study, the doses (0.02-4 mg/kg) was chosen according to the contents of some raw materials in previous reports and our preliminary experiment.

This study was conducted to investigate the potential hazards of BDE-209 on growth, meat quality and blood profiles (hematological, biochemical and anti-oxidant parameters) in chicken. Because the doses of BDE-209 (approach to the real concentrations in animal feeds) used in this study are much lower than the mortality dose, there is only one dead chicken in the control and BDE-209 treated group at the dose of 0.4 mg/kg throughout the 6-week experiment, and it was unrelated to BDE-209 treatment. We have described in the revised manuscript.

Comment:

For statistical analyses, since you data for monitoring growth and feed intake follow a normal distribution it would be more interesting to use a statistical model (General linear model or Mixed if you have individually identified the chickens) to take into account for these growth variables the effects of age and sex. Similarly for the other parameters studied, take into account the effect of sex as it is a factor that can significantly influence the results.

Response:

Thanks for your good suggestion. According to the comments, we have performed a regression analysis as well GLM using the method of least-significant difference (LSD) multiple comparisons. We are very sorry that we have not described clearly in the section of materials and methods. As description above, only male broilers were chosen in this study and we have described it in the revised manuscript.

Comment:

Discussion did not flow with data presented. It is necessary to explain the results observed and to check if the observed values are normal (physiological and zootechnical) or not. Also give the mortality rates and explain the degree of toxicity of the supplemented feeds.

Response:

    Thanks for your comments and good suggestion. We have amended in the revised manuscript. As description above, the doses of BDE-209 (approach to the real concentrations in animal feeds) used in this study are much lower than the mortality dose, there is only one dead chicken in the control and 0.4 mg/kg BDE-209 group throughout the 6-week experiment, and it was unrelated to BDE-209 treatment. We have described in the revised manuscript.

Comment:

Conclusion: Adapt your conclusion to the results of mortality and sex ratio by group.

Response:

Thanks for your comments and good suggestion. As description above, only male broilers were chosen in this study and we have described it in the revised manuscript. Please see the revised manuscript.

Reviewer 3 Report

The paper reads well, was easy to understand and follow. I thoroughly enjoyed reading your paper.

I have few comments as below:

Summary and abstract:

Could you please add information regarding the dose rate, if the lowest dose was detrimental?

Materials and methods: 

The materials and methods are not clear. Please rewrite this section. Please include how many pens were there, how many replicates per treatment and how many birds per pen/cage.

Line 77: It is not clear if the body weight was used to maintain the uniformity of the treatment groups or they were divided into the groups according to their body weight. Please make it clear.

Line 82: The sentence is not clear whether it means all conditions were same except diet with different levels of BDE-209. Please rewrite it.

Line 88: What do you mean by cervical blood letting? Does that mean cervical dislocation and jugular vein cut? Please clarify.

Line 115: Remove extra bracket after GLM.

Results:

Line 150: Correct bellow to 'below'.

Line 151: Significant difference; Is it between columns or rows?

Conclusion:

Add details about the dose rate, its effect.

Line 249: correct affected to "effects"

Author Response

The paper reads well, was easy to understand and follow. I thoroughly enjoyed reading your paper.

I have few comments as below:

Comment:

Summary and abstract:

Could you please add information regarding the dose rate, if the lowest dose was detrimental?

Response:

Thanks for your comments and good suggestions. The doses used in this study are lower than previous studies and the lowest dose is approach to the maximal concentration in animal feeds and raw materials. We have amended in the revised manuscript.

Comment:

Materials and methods:

The materials and methods are not clear. Please rewrite this section. Please include how many pens were there, how many replicates per treatment and how many birds per pen/cage.

Line 77: It is not clear if the body weight was used to maintain the uniformity of the treatment groups or they were divided into the groups according to their body weight. Please make it clear.

Line 82: The sentence is not clear whether it means all conditions were same except diet with different levels of BDE-209. Please rewrite it.

Line 88: What do you mean by cervical blood letting? Does that mean cervical dislocation and jugular vein cut? Please clarify.

Line 115: Remove extra bracket after GLM.

Results:

Line 150: Correct bellow to 'below'.

Line 151: Significant difference; Is it between columns or rows?

Response:

Thanks for your comments and good suggestion. In this study, 80 male broilers were randomly allocated into 4 groups according to their body weights after feeding with the basal corn-soybean diet for one week, 20 broilers in each group, 5 replicate cages of 4 broilers per cage. The Materials and Methods is not detailed and clear in our previous manuscript and we have amended this section and corrected our mistakes point to point according to the comments in the revision. Please see the corrections in the revised manuscript.

Comment:

Conclusion:

Add details about the dose rate, its effect.

Line 249: correct affected to "effects"

Response:

Thanks for your good suggestion and we have made these corrections in the revised manuscript.

Reviewer 4 Report

Congratulations to authors for a novel and interesting paper. However, a minor revision that authors should be considered to address, as following:

1. Based on what the authors included BDE-209 at 0.02, 0.4 and 4  mg/kg? 

2. The quality of figures were very poor, and the authors need to improve the resolution of photos.

3. In all tables, pooled SEM should be used instead of individual SEM for each treatment.

4. There were 4 levels of BDE-209 included, the authors should do regression analysis.

Author Response

Congratulations to authors for a novel and interesting paper. However, a minor revision that authors should be considered to address, as following: 

Comment:

  1. Based on what the authors included BDE-209 at 0.02, 0.4 and 4  mg/kg? 

Response:

Thanks for your comments and good suggestions.

Previous retroactive studies suggested that BDE-209 has been detected in the commercial feeds of livestock (including pigs, chickens, ducks, fish, etc.) and raw materials (including fish meal, soybean, rapeseeds, etc.) though the concentrations are very low and varied very greatly (ranged from 0.3-20 ng/g for common animal feeds and raw materials).

The tolerance to BDE-209 of animals varied greatly and the mortality dose is much higher than those used in the studies. Higher doses in diets almost were chosen (ranged from 1-85 mg/kg) in previous studies for toxicity assessment. In this study, the doses (0.02-4 mg/kg) was chosen according to the real contents of some raw materials in previous reports and our preliminary experiment.

Comment:

  1. The quality of figures were very poor, and the authors need to improve the resolution of photos.

Response:

Thanks for your good suggestion and we have improved the resolution of pothos in the revised manuscript.

Comment:

  1. In all tables, pooled SEM should be used instead of individual SEM for each treatment.

Response:

Thanks for your good suggestion. We have presented the pooled SEM instead of individual SEM for each treatment in the revision.

Comment:

  1. There were 4 levels of BDE-209 included, the authors should do regression analysis.

Response:

Thanks very much for your good suggestion. In the revised manuscript, we have performed a regression analysis with SPSS as well GLM procedure using the method of least-significant difference (LSD) multiple comparisons.

Round 2

Reviewer 1 Report

The manuscript was significantly improved and everything is more clear now. However, the paper is plagued with errors, typos and bad grammar and writing style, for example "underlying toxic mechanism of BDE-209 both in livestockhuman beings and animals. " or the part discussing "human beings at the top of the food chain" etc. The manuscript now needs detailed English editing.

Author Response

Comments:

The manuscript was significantly improved and everything is more clear now. However, the paper is plagued with errors, typos and bad grammar and writing style, for example "underlying toxic mechanism of BDE-209 both in livestockhuman beings and animals. " or the part discussing "human beings at the top of the food chain" etc. The manuscript now needs detailed English editing.

Respose:

Thanks for your consideration and good suggestions. Because too much modification trace in this revision might confuse editors and reviewers, we have accepted all the editing in the last revision in Word, and submitted it for English editing in MDPI system (English-27014). Besides, some minor corrections have been made in the revised version, such as adding a reference in the method part. Thanks very much for your professional suggestions to improve our manuscript.

Reviewer 2 Report

Thank you for making thoughtful and significant improvements to the manuscript. The focus and readability are greatly improved.  The manuscript is also much more focused and the aims are clearer than in the prior submission.

Author Response

Comments:

Thank you for making thoughtful and significant improvements to the manuscript. The focus and readability are greatly improved.  The manuscript is also much more focused and the aims are clearer than in the prior submission.

Respose:

Thanks for your consideration and good suggestions. Because too much modification trace in this revision might confuse editors and reviewers, we have accepted all the editing in the last revision in Word, and submitted it for English editing in MDPI system (English-27014). Besides, some minor corrections have been made in the revised version, such as adding a reference in the method part. Thanks very much for your professional suggestions to improve our manuscript.